# Silica-grafted ionic liquids for revealing the respective charging behaviors of cations and anions in supercapacitors

Qingyun Dou[1,2], Lingyang Liu[1,2], Bingjun Yang[1], Junwei Lang[1] & Xingbin Yan [1]

Supercapacitors based on activated carbon electrodes and ionic liquids as electrolytes are capable of storing charge through the electrosorption of ions on porous carbons and represent important energy storage devices with high power delivery/uptake. Various computational and instrumental methods have been developed to understand the ion storage behavior, however, techniques that can probe various cations and anions of ionic liquids separately remain lacking. Here, we report an approach to monitoring cations and anions independently by using silica nanoparticle-grafted ionic liquids, in which ions attaching to silica nanoparticle cannot access activated carbon pores upon charging, whereas free counter-ions can. Aided by this strategy, conventional electrochemical characterizations allow the direct measurement of the respective capacitance contributions and acting potential windows of different ions. Moreover, coupled with electrochemical quartz crystal microbalance, this method can provide unprecedented insight into the underlying electrochemistry.

---

[1] Laboratory of Clean Energy Chemistry and Materials, State Key Laboratory of Solid Lubrication, Lanzhou Institute of Chemical Physics, Lanzhou 730000 China. [2] University of Chinese Academy of Sciences, Beijing 100080 China. Correspondence and requests for materials should be addressed to X.Y. (email: xbyan@licp.cas.cn)

Electrochemical double-layer capacitors (EDLCs, also called supercapacitors) based on porous activated carbons (ACs) are promising energy storage devices owing to their high power density, fast charge/discharge capability, and ultra-long cycle life[1–7]. In EDLCs, electrolytes, like AC electrodes, are useful in directly determining the overall performance of the devices[7–12]. The calculation formula of energy density ($E = 1/2\ CV^2$, where $E$ is the energy density, $C$ is the specific capacitance, and $V$ is the operating potential window) indicates that the $E$ of the device greatly depends on the $V$ of electrolytes. Ionic liquids (ILs) have wider operating potential windows, lower vapor pressures, and better chemical and thermal stability than aqueous electrolytes and are thus frequently used as non-aqueous electrolytes[9–13]. Furthermore, ILs exhibit remarkably increased energy densities and wide applicable temperature ranges; thus, they are considered promising electrolytes for EDLCs[8–11,14,15].

ILs are composed of organic cations and organic or inorganic anions, and numerous cations and anions can be selected for IL combinations[10,16,17]. These cations and anions have distinctly asymmetric chemical structures, and thus the positive and negative electrodes of EDLCs in IL electrolytes exhibit uneven working potential distributions and variable electrochemical behaviors[14,18–21]. Furthermore, the charge storage capacities of ACs in EDLCs can be directly affected by the types and molecular structures of cations and anions of ILs[14,19,20]. Therefore, studying the effects of the intrinsic structures of cations and anions on the capacitive properties of ACs and determining the real charge storage mechanisms of EDLCs in IL electrolytes are necessary for the proper selection of ILs and rational design of high-performance EDLCs[22,23].

Many computational simulations and instrumental analysis methods have been developed for the characterization of the storage mechanisms of ACs of EDLCs in ILs. The charge storage behaviors of AC electrodes in ILs can be understood from the theoretical level through computational simulations; the capacitance and ion dynamics of EDLCs strongly depend on factors, such as relative pore or ion size[19,24–26], confinement effect[26–28], and ionophilic/ionophobic pores[28–30]. Simulations must undergo necessary simplifications such that simulation time and accuracy are effectively balanced. Consequently, they usually do not include all details of complex porous/and electronic structures of ACs at experimental conditions. By contrast, reported instrumental analysis methods using infrared (IR) spectroscopy[31,32], small-angle neutron scattering (SANS)[33–35], electrochemical quartz crystal microbalance (EQCM)[36–44], and nuclear magnetic resonance (NMR)[44–50] are effective techniques and can be effectively used for the observation of variations in ion concentration in AC pores during charging or discharging; additionally, purely perm-selective behaviors (adsorption of counter-ions, which have an electric charge opposite to that of electrodes) and ion swapping behaviors (adsorption of counter-ions, together with the desorption of co-ions, which have the same electric charge as that of electrodes) occur during the polarization of AC electrodes[22,23]. Cations or anions of an IL can be individually monitored through an IR technique because each type of ions has its own characteristic signal in the IR spectrum. Meanwhile, SANS can be used for the direct characterization of ions in differently sized pores, and the dynamic capability of EQCM allows real-time measuring of changes in electrode mass during charging. NMR is an effective detection technique with excellent elemental selectivity. The peak area in an NMR spectrum is proportional to the corresponding ion concentration. Moreover, NMR can be used for the selectively monitoring of in-pore ions with resonances shifting to a frequency lower than that of ex-pore ions. Therefore, in-pore cations and anions can be separately characterized and their concentrations can be fully quantified through NMR. Nevertheless, IR, SANS, EQCM, and NMR require specialized and expensive instruments, and have their respective analytical limitations[22]. Thus, for the in-depth understanding of the energy storage mechanisms of EDLCs, universal experimental methods that can characterize the influences of various cations and anions of ILs on the charge storage of ACs need to be developed.

EDLCs use porous ACs with high specific surface area and uniform pore size distribution. When only one type of ion (either cation or anion) of an IL is allowed to enter carbon pores, the respective electrochemical behaviors of cations and anions in carbon pores can be analyzed. To this end, we develop an organic synthesis strategy for silica nanoparticle-grafted ILs to realize this idea (Fig. 1). In these synthetic SiO$_2$-grafted ILs, only one ion is free, and the ion used for balancing the charge is directly attached to silica nanoparticles. Moreover, ACs are selected as study samples. Their pore sizes are less than the diameter of silica nanoparticles. We find that, when the SiO$_2$-grafted ILs are used as electrolytes for AC electrodes, the ions covalently attached to silica nanoparticles are restricted to access to the AC pores, and only the free ions are able to enter the pores during charging or discharging (Fig. 1d). Thus, free ions in pores can be quantitatively analyzed through conventional electrochemical tests. By using AC electrodes, we are able to directly measure the capacitance contributed by cations and anions, and we find that each ion is adsorbed or desorbed within its own specific potential window during charge storage. Combination of the measurement results of respective charging behaviors of cations and anions with the results of EQCM investigation implies that purely perm-selective behavior, which is determined by the nature of ions, is dominant at a high polarized window, and only counter-ion adsorption contributes capacitance; by contrast, perm selectivity and ion swapping both occur at a low polarized window, which is determined by the competition between counter-ions and co-ions.

## Results

**Material synthesis and characterization.** Four types of SiO$_2$-grafted ILs were synthesized through the aforementioned organic synthesis methods. Methyl imidazolium cation (MIM$^+$)[51,52] and trifluoromethanesulfonyl imide anion (NTf$^-$) were covalently attached to 7 nm commercial silica nanoparticles (LUDOX SM-30 colloidal silica, Aldrich). Meanwhile, MIM$^+$ or NTf$^-$ possesses an ionic moiety that balances the charge of free ions and can bear a carbon chain that can attach to SiO$_2$ nanoparticles. Synthetic SiO$_2$-MIM$^+$ was combined either with bis(trifluoromethanesulfonyl) imide (NTf$_2^-$) or hexafluorophosphate (PF$_6^-$), and synthetic SiO$_2$-NTf$^-$ was combined with either 1-butyl-3-methylimidazolium (BMIM$^+$) and tetrabutylammonium (NBu$_4^+$) (Fig. 1a).

Figure 2 shows the reaction steps for synthesizing SiO$_2$-grafted ILs, namely, SiO$_2$-IL-NTf$_2$ (**1**), SiO$_2$-IL-PF$_6$ (**2**), SiO$_2$-IL-BMIM (**3**), and SiO$_2$-IL-NBu$_4$ (**4**) (see methods). The main issue of this synthetic method was the preparation of key intermediate products, particularly SiO$_2$-MIM-Cl (**5**) and SiO$_2$-NTf-Li (**6**) for the attachment of ions (MIM$^+$ and NTf$^-$) to SiO$_2$ nanoparticles. To obtain compound **9**, we actuated a quaternization reaction between (3-chloropropyl)trimethoxysilane (**7**) and 1-methylimidazole (**8**). The produced compound **9** was subsequently attached to the SiO$_2$ nanoparticles by a condensation reaction for the production of SiO$_2$-MIM-Cl (**5**). Compound **5** was subjected to an ion exchange with LiNTf$_2$ (**10**) and LiPF$_6$ (**11**) and successfully converted into SiO$_2$-IL-NTf$_2$ (**1**) and SiO$_2$-IL-PF$_6$ (**2**), respectively. For the preparation of SiO$_2$-NTf-Li (**6**), compound **12** was first converted into compound **13** via trifluoromethanesulfonylation, followed by lithiation. Then, compound **13** was attached to the SiO$_2$ nanoparticles for SiO$_2$-

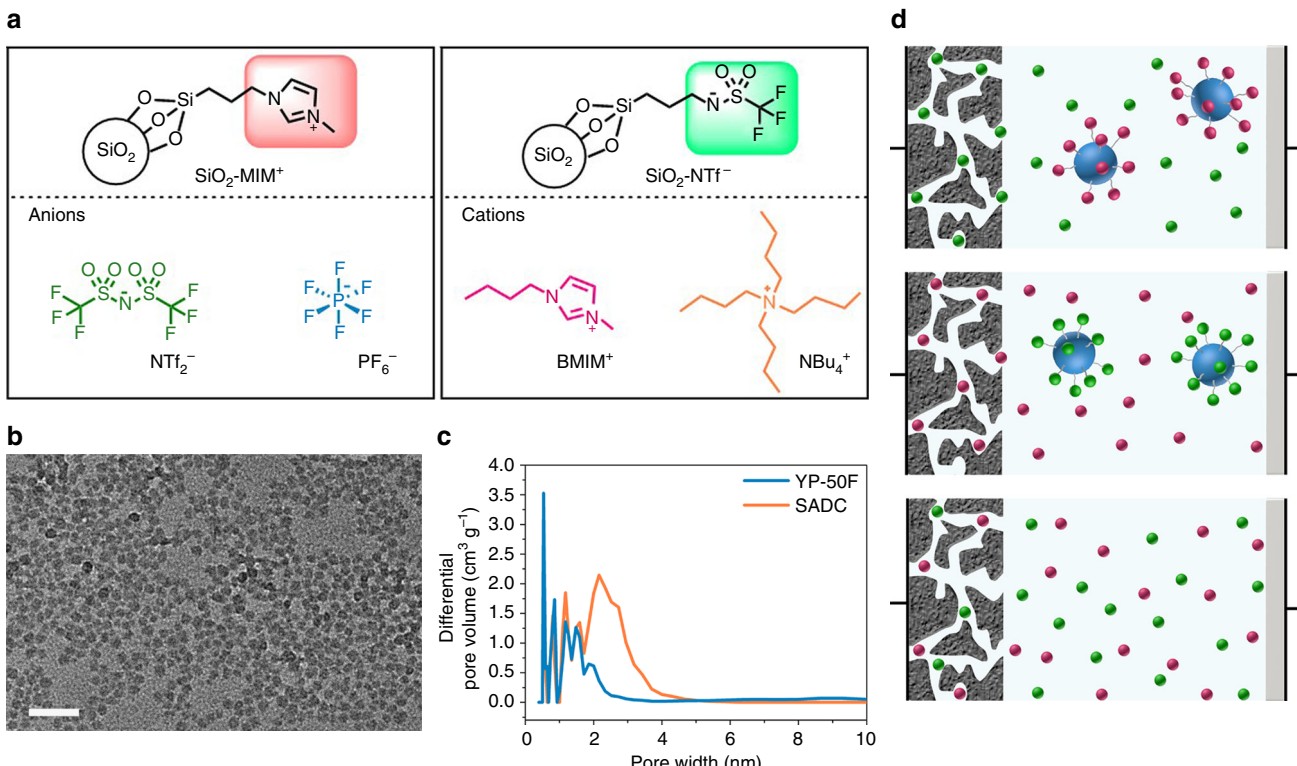

**Fig. 1** Schematic of the strategy for respectively characterizing the charging behavior of cations and anions of ionic liquids in electrochemical double-layer capacitors (EDLCs). **a** Structures of cations, anions, and silica nanoparticle-grafted ions. **b** Transmission electron micrograph of silica nanoparticles with a uniform diameter of 7 nm. Scale bar, 50 nm. **c** Pore size distributions of two activated carbons, which indicate that the pore sizes in two activated carbons are smaller than the diameter of silica nanoparticles. **d** Schematic of charge storage of activated carbon electrodes during polarization in different types of ionic liquids: silica nanoparticle-grafted cations and free anions (top), silica nanoparticle-grafted anions and free cations (middle), conventional cations and anions (bottom). Big blue balls represent silica nanoparticles, small wine balls represent cations, and green balls represent anions

NTf-Li (**6**) production. Lithium ion (Li$^+$) in SiO$_2$-NTf-Li (**6**) was substituted with BMIM$^+$ (**14**) or NBu$_4^+$ (**15**) through ion exchange for SiO$_2$-IL-BMIM (**3**) and SiO$_2$-IL-NBu$_4$ (**4**) production.

The chemical structures and purity of the synthesized SiO$_2$-grafted ILs were confirmed by $^1$H NMR and $^{19}$F NMR. The results indicated that MIM$^+$ and NTf$^-$ were successfully attached to the SiO$_2$ nanoparticles (Supplementary Figs. 3–8). Energy-dispersive X-ray spectroscopy (Supplementary Fig. 13) and inductively coupled plasma optical emission spectroscopy (see Supplementary Information) results verified that the absence of Cl, Br, or Li in the SiO$_2$-grafted ILs, suggesting that excess salt (LiCl or LiBr) was completely removed after ion exchange. The above measurements confirmed that only one type of ion was free in each SiO$_2$-grafted IL. The IR spectra further demonstrated the existence of characteristic bands for the SiO$_2$-grafted ILs (Supplementary Fig. 14), indicating the successful grafting of IL molecules on the SiO$_2$ nanoparticles. The ion concentrations (the ratio of the mole number of ions to the mass of the corresponding SiO$_2$-grafted IL) determined by an internal standard method (Supplementary Figs. 9–12) in the four SiO$_2$-grafted ILs were 0.93, 0.95, 0.86, and 0.67 mmol g$^{-1}$. Thermogravimetric analysis results indicated that the SiO$_2$-grafted ILs were thermally stable up to 200 °C and their mass losses at 800 °C were 49.6, 35.1, 37.4, and 40.6 wt% (Supplementary Fig. 15).

**Electrochemical measurements.** A commercially available AC (YP-50F, Kuraray Chemical, Japan) was used as electrode (Supplementary Fig. 1). Four synthesized SiO$_2$-grafted ILs and the corresponding ILs or organic salts (BMIM-NTf$_2$, BMIM-PF$_6$, NBu$_4$-NTf$_2$, and NBu$_4$-PF$_6$) dissolved in anhydrous propylene carbonate (PC) with the same concentration of 0.25 M were used as electrolytes (Supplementary Fig. 16). The electrochemical behavior of AC electrodes in electrolytes was characterized in a closed three-electrode system. Figure 3 shows the cyclic voltammetry (CV) curves of the YP-50F electrode in four SiO$_2$-grafted IL electrolytes at the same scan rate (5 mV s$^{-1}$). The CV curves of SiO$_2$-IL-BMIM within different operating potential windows (Fig. 3a), which should actually exhibit the sorption property of BMIM$^+$ because only it can freely enter the pores of YP-50F, all showed that the specific capacitance rapidly decreased when the electrode was positively polarized to ~0.9 V/ref (the maximum slope was around this potential). This rapid decrease in capacitance indicated that the upper acting potential for the capacitance contribution of BMIM$^+$ was ~0.9 V/ref (Supplementary Note 1, Supplementary Figs. 17 and 18). The CV curves of SiO$_2$-IL-NBu$_4$ exhibited a similar feature (Fig. 3c). Thus, the upper acting potential for the capacitance contribution of NBu$_4^+$ was ~1.0 V/ref. The rapid decrease in capacitance was also observed in the CV curves of SiO$_2$-IL-NTf$_2$ (Fig. 3b) and SiO$_2$-IL-PF$_6$ (Fig. 3d), but the decrease occurred at negatively polarized directions (−0.1 V/ref for SiO$_2$-IL-NTf$_2$ and 0.1 V/ref for SiO$_2$-IL-PF$_6$). The lower acting potentials for the capacitance contributions of anions were ~−0.1 V/ref for NTf$_2^-$ and ~0.1 V/ref for PF$_6^-$. Comparisons between the CV curves of two cations with the same operating potential windows and between those of two anions were conducted. The results showed that the capacitance contribution of BMIM$^+$ was larger than that of NBu$_4^+$ for the YP-50F electrode, and that of NTf$_2^-$ was larger than that of PF$_6^-$ (Supplementary

**Fig. 2** Synthesis of four silica nanoparticle-grafted ionic liquids (SiO$_2$-grafted ILs). Methyl imidazolium cation (MIM$^+$) and trifluoromethanesulfonyl imide anion (NTf$^-$) were firstly covalently grafted to silica nanoparticles to synthesize the key intermediate products SiO$_2$-MIM$^+$ and SiO$_2$-NTf$^-$. Synthetic SiO$_2$-MIM$^+$ was combined either with bis(trifluoromethanesulfonyl) imide (NTf$_2^-$) or hexafluorophosphate (PF$_6^-$), and synthetic SiO$_2$-NTf$^-$ was combined with either 1-butyl-3-methylimidazolium (BMIM$^+$) and tetrabutylammonium (NBu$_4^+$) through ion exchange to obtain SiO$_2$-IL-NTf$_2$ (**1**), SiO$_2$-IL-PF$_6$ (**2**), SiO$_2$-IL-BMIM (**3**), and SiO$_2$-IL-NBu$_4$ (**4**)

Fig. 19). The difference may be attributed to the differences between the compared ions with respect to size and chemical structure, which may have affected the interface interaction between the ions and pore surfaces of carbon electrodes[14,18]. Meanwhile, some studies reported that one type ions can solely contribute to charge storage mechanisms[39,40]. The CV curves of SiO$_2$-IL-NTf$_2$ and SiO$_2$-IL-PF$_6$ were similar to that of another commercial AC (YP-17, Kuraray Chemical, Japan) electrode in tetraoctylammonium tetrafluoroborate (TOA-BF$_4$) IL[39], in which the charge at the negatively polarized electrode rapidly decreases because the bulky 1.12 nm TOA$^+$ cations are inaccessible to most of the carbon pores. Basing on this similarity, we concluded that ions attached to SiO$_2$ nanoparticles were indeed restricted out of the pores of YP-50F during the charging process.

We summed the CV curves of SiO$_2$-IL-BMIM and SiO$_2$-IL-NTf$_2$, SiO$_2$-IL-BMIM and SiO$_2$-IL-PF$_6$, SiO$_2$-IL-NBu$_4$ and SiO$_2$-IL-NTf$_2$, and SiO$_2$-IL-NBu$_4$ and SiO$_2$-IL-PF$_6$ to obtain the summed CV curves of the YP-50F electrode in four organic electrolytes (BMIM-NTf$_2$, BMIM-PF$_6$, NBu$_4$-NTf$_2$, and NBu$_4$-PF$_6$), as shown in Fig. 4a–d. The summed CV curves (blue dotted lines) represented a perfect state, in which all cations and anions fully contributed their capacitance. On the two sides of each highly polarized region, a purely adsorptive mechanism should be observed. In particular, only the counter-ions contributed capacitance (light red, orange, blue, and green regions). Taking Fig. 4a as an example, the capacitance only originated from the contribution of BMIM$^+$ at a negatively polarized potential window (below −0.1 V/ref); and at the high positively polarized

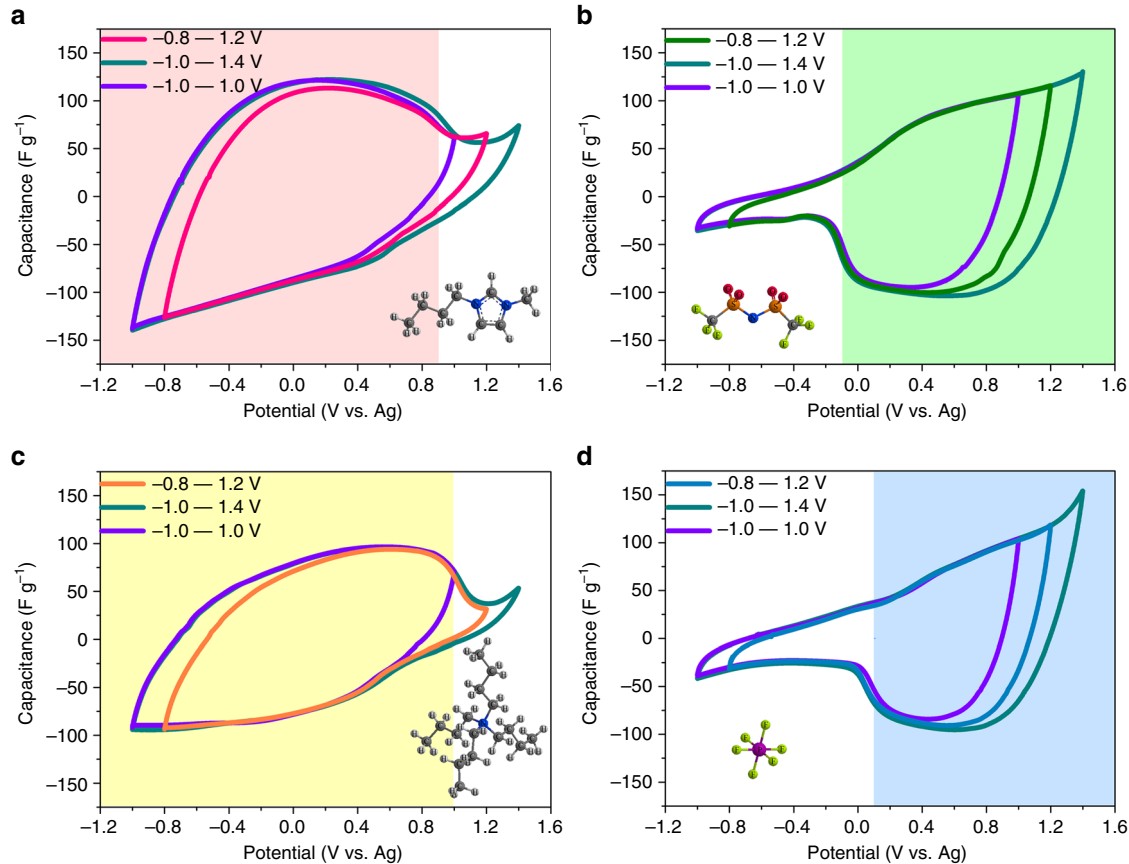

**Fig. 3** Cyclic voltammetry (CV) curves of YP-50F electrode in four SiO$_2$-grafted ILs with the same concentration of 0.25 M in propylene carbonate (PC). **a** SiO$_2$-IL-BMIM, **b** SiO$_2$-IL-NTf$_2$, **c** SiO$_2$-IL-NBu$_4$, and **d** SiO$_2$-IL-PF$_6$. The curves were scanned within different operating potential windows with the same scan rate of 5 mV s$^{-1}$

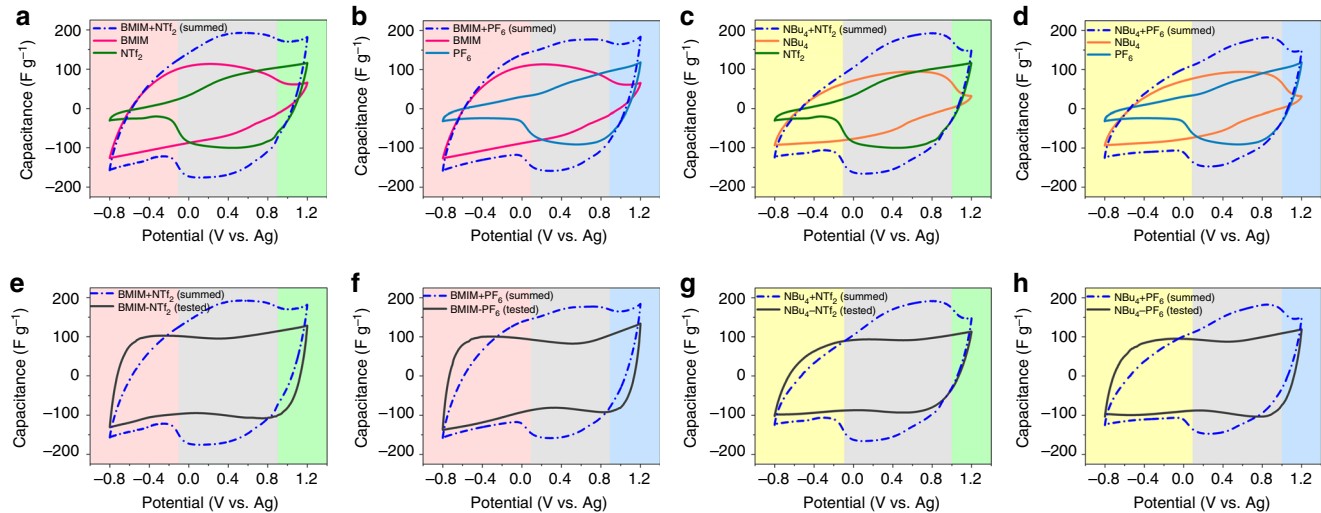

**Fig. 4** Comparison of the summed CV curves with the real CV curves. Summed cyclic voltammetry (CV) curves of YP-50F electrode in four electrolytes by summing the CV curves of YP-50F electrode in four SiO$_2$-grafted ILs shown in Fig. 3: **a** SiO$_2$-IL-BMIM and SiO$_2$-IL-NTf$_2$, **b** SiO$_2$-IL-BMIM and SiO$_2$-IL-PF$_6$, **c** SiO$_2$-IL-NBu$_4$ and SiO$_2$-IL-NTf$_2$, and **d** SiO$_2$-IL-NBu$_4$ and SiO$_2$-IL-PF$_6$. Summed CV curves and real CV curves of YP-50F electrode in four corresponding conventional organic electrolytes: **e** BMIM-NTf$_2$, **f** BMIM-PF$_6$, **g** NBu$_4$-NTf$_2$, and **h** NBu$_4$-PF$_6$. All tested CV curves were scanned within the operating potential window of −0.8 V to 1.2 V/ref with the same scan rate of 5 mV s$^{-1}$. The organic electrolytes had the same concentration of 0.25 M in propylene carbonate (PC)

potential window (above 0.9 V/ref) the capacitance was only contributed by NTf$_2^-$. At a middle potential window of −0.1 to 0.9 V/ref (gray region), both BMIM$^+$ and NTf$_2^-$ presented charge storage, and the capacitance value of the summed CV curve was

larger than that of the CV curve in SiO$_2$-IL-BMIM or SiO$_2$-IL-NTf$_2$. However, is it true? To determine whether the two ions at the middle region can promote each other and thus exhibit summed capacitance, we tested the actual CV curves of the four

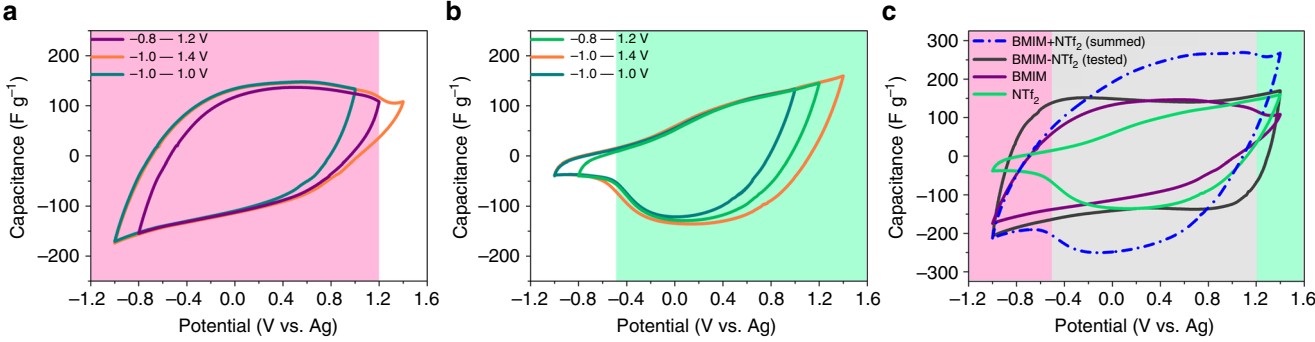

**Fig. 5** Electrochemical characterization of sodium alginate-derived activated carbon (SADC) electrode. Cyclic voltammetry (CV) curves in **a** SiO$_2$-IL-BMIM and **b** SiO$_2$-IL-NTf$_2$ with the same concentration of 0.25 M in propylene carbonate (PC). The curves were scanned within different operating potential windows with the same scan rate of 5 mV s$^{-1}$. **c** Comparison between the summed and real CV curves of SADC electrode in BMIM-NTf$_2$ at an operating potential window of −1.0 to 1.4 V/ref

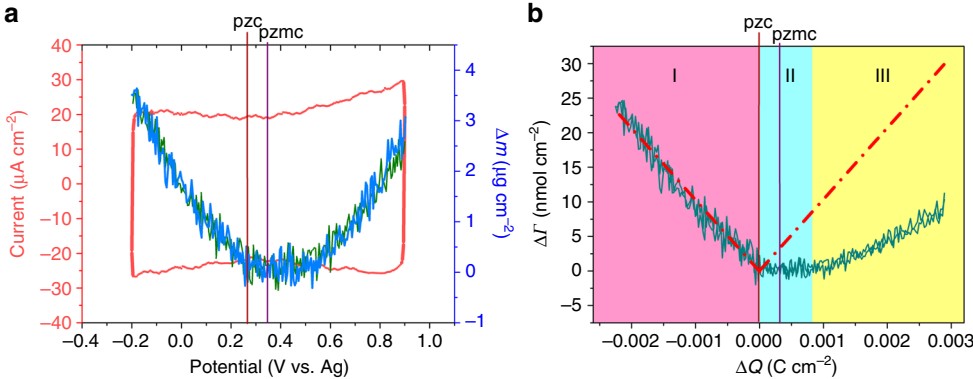

**Fig. 6** Electrochemical quartz crystal microbalance (EQCM) analyses of YP-50F-coated quartz electrode in 0.25 M BMIM-NTf$_2$/PC electrolyte. **a** Cyclic voltammetry (CV) curve with a scan rate of 5 mV s$^{-1}$ and the corresponding EQCM mass change. **b** Ion population change as a function of charge density during polarization. Cyan solid line represents the experimental ion population change. Red dashed line is the theoretical ion population change based on perm-selective behavior

conventional ILs, which all exhibited a nearly rectangular shape (black lines in Fig. 4e–h), indicating the typical EDLC behavior of AC electrode. The summed curves were compared with actual curves (Fig. 4e–h), and the results showed that the capacitance values of the summed CV curves were significantly larger than that of the actual CV curves at low polarized regions (middle gray regions, defined on the basis of the acting potential of each SiO$_2$-grafted IL shown in Fig. 3). The summed CV curves represented the ideal sum of the respective sorption contribution of cations and anions on carbon surfaces, and the actual CV curves represented the real overall sorption behavior of cations and anions on the same carbon surfaces. Therefore, the comparison results indicated that the cations and anions competed with each other within the middle region. In addition, the capacitance of the summed curves was smaller than that of the conventional electrolytes upon the reversal of the potential possibly because of the decreased ionic conductivity of the SiO$_2$-grafted ILs (Supplementary Table 1). The decrease may be caused by the restricted movement of the free ions of the SiO$_2$-grafted ILs because of the nearly immobile SiO$_2$ nanoparticles.

After separately monitoring the charging behavior of cations and anions of ILs using YP-50F electrode as the study subject, we further verified the feasibility of this strategy for other ACs. The homemade sodium alginate-derived AC (denoted as SADC) was selected as study sample. Figure 5a, b show the CV curves of SADC in SiO$_2$-IL-BMIM and SiO$_2$-IL-NTf$_2$ ILs, both of which had the same concentration of 0.25 M in PC. Figure 5c shows the summed and real CV curves of the SADC in the BMIM-NTf$_2$/PC

electrolyte. Similar CV shapes appeared for the SADC electrode. BMIM$^+$ dominated at the potential below −0.5 V/ref, NTf$_2^-$ dominated at the potential above 1.2 V/ref, and BMIM$^+$ and NTf$_2^-$ competed at the middle potential window of −0.5 to 1.2 V/ref. The results of the CV comparison between the YP-50F and SADC electrodes (Supplementary Fig. 20) indicated that latter had higher capacitance than former in the BMIM-NTf$_2$ electrolyte system. It because the SADC had higher Brunauer–Emmett–Teller surface area than the YP-50F (Supplementary Figs. 1 and 2).

As mentioned before, changes in electrode mass during dynamic ion charging can be measured in real time through EQCM[36–44]. Thus, in situ EQCM investigation was conducted to study the specific charge mechanisms of cations and anions within the competitive potential window indicated above. Figure 6a shows the CV curve and the simultaneous mass change ($\Delta m$; calculated from the frequency response change of EQCM experiment through Sauerbrey's equation (see Supplementary Information)) of YP-50F-coated quartz (as working electrode; Supplementary Note 2, Supplementary Fig. 22) in a BMIM-NTf$_2$/PC electrolyte within an operating potential window of −0.2 to 0.9 V/ref. Potential window was selected by a chronoamperometry test[44] (Supplementary Note 3, Supplementary Figs. 23 and 24). Moreover, when the YP-50F-coated quartz electrode was cycled between −0.2 V and 0.9 V/ref, the resonance width change ($\Delta W$) should be much smaller than the simultaneous resonance frequency change ($\Delta f$; Supplementary Note 4, Supplementary Fig. 25)[37,41]. In Fig. 6a, at potential scans ranging from 0.9 V/ref

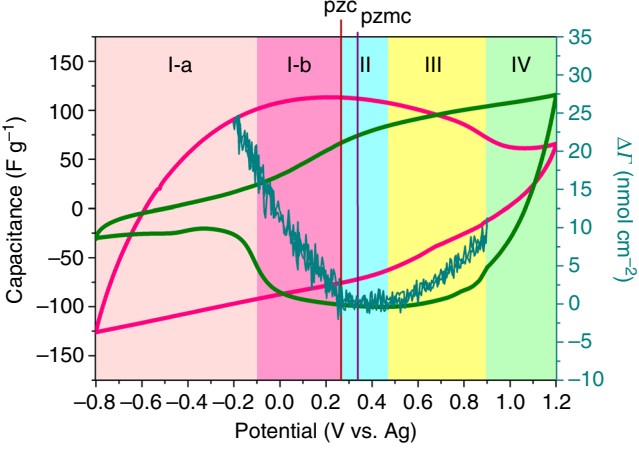

**Fig. 7** Combination of electrochemical quartz crystal microbalance (EQCM) data with the cyclic voltammetry (CV) curves shown in Fig. 4a

to −0.2 V/ref, the mass change (olive line) typically showed a V shape. At the subsequent potential back scanning from −0.2 V/ref to 0.9 V/ref, the mass change (blue line) was nearly coincidental with the previous one, indicating the reversibility of the current experiment. The potential of zero mass change (pzmc), which corresponded to the minimum mass change, was close to 0.34 V/ref. The evaluated potential of zero charge (pzc) of 0.27 V/ref was shifted to the left with respect to pzmc (Supplementary Fig. 21).

Figure 6b shows that the ion population change ($\Delta\Gamma$) is a function of charge density ($\Delta Q$; calculated by integration of the related CV). The cyan line represents the experimental ion population change obtained by dividing the mass change by the molecular weight of the ion, and the red dashed line is the theoretical ion population change calculated by assuming adsorption of solely counter-ions with the use of Faraday's law (see Supplementary Information). During the negative polarization from pzc ($\Delta Q < 0$), the experimental ion population change was nearly equal to the theoretical one, implying that the charge on the negatively polarized electrode was solely balanced by the principle adsorption of counter-ions ($BMIM^+$), and no solvent was involved[37,38,43] (Fig. 6b, domain I). During the positive polarization from pzc ($\Delta Q > 0$), the experimental ion population change exhibited a flat minimum close to pzmc at a low charge density (Fig. 6b, domain II) and demonstrated an ion swapping behavior, that is, co-ion ($BMIM^+$) desorption and counter-ion ($NTf_2^-$) adsorption were both involved in the charge storage mechanism[37,43]. The shift of pzc to the left with respect to pzmc indicated that desorption of co-ions ($BMIM^+$) played a dominant role in the potential between pzc and pzmc[42]. At a high charge density (Fig. 6b, domain III), changes in the experimental ion population continuously increased, and the slope gradually approached the theoretical one, illustrating that the role of co-ion desorption diminished, whereas counter-ions played a dominant role in the charge storage process[42,43].

The EQCM results were further combined with the results obtained in the CV curves of SiO₂-grdfted ILs (SiO₂-IL-BMIM (red curve), SiO₂-IL-NTf₂ (green curve)) to yield an in-depth understanding of the charging mechanism of carbon YP-50F in BMIM-NTf₂/PC (Fig. 7). At highly negatively and positively polarized regions (below −0.1 V/ref and above 0.9 V/ref), only perm-selective behavior was determined because solely one type of two ions ($BMIM^+$ or $NTf_2^-$) exhibited charging behaviors in the two regions. Within the intermediate potential window of −0.1 to 0.9 V/ref, perm selectivity and ion swapping both occurred and were determined by the competition between cations ($BMIM^+$) and anions ($NTf_2^-$). Specifically, at negative

polarization (the potential was below pzc), solely perm-selective behavior was observed. In domain I-a (potential below −0.1 V/ref), the comparison of CV curves of SiO₂-IL-BMIM and SiO₂-IL-NTf₂ indicated that the perm-selective behavior was determined by the intrinsic characteristic of ions, and the counter-ions ($BMIM^+$) alone stored charge, whereas co-ions ($NTf_2^-$) did not. In domain I-b (potential from −0.1 V/ref to 0.27 V/ref), counter-ions ($BMIM^+$) and co-ions ($NTf_2^-$) presented competition, and the EQCM result demonstrated that the capacitance only originated from the adsorption of counter-ions ($BMIM^+$). At positive polarization (the potential was above pzc), the EQCM experiment revealed that at the low positively polarized region (0.27 V/ref to 0.9 V/ref, domains II and III), the adsorption of counter-ions ($NTf_2^-$) and desorption of co-ions ($BMIM^+$) both contributed capacitance, and the role of counter-ion ($NTf_2^-$) adsorption became evident as the polarization degree increased. At the highly positively polarized region (above 0.9 V/ref, domain IV), only the adsorption of the counter-ions ($NTf_2^-$) was involved.

## Discussion

In summary, we demonstrated a method to separately monitor cations ($BMIM^+$, $NBu_4^+$) and anions ($NTf_2^-$, $PF_6^-$) in AC pores by utilizing four SiO₂-grafted ILs, which were successfully prepared by organic synthesis. The charging behaviors of $BMIM^+$, $NBu_4^+$, $NTf_2^-$, and $PF_6^-$ at different potential states can be directly measured through conventional CV tests and by ensuring that only one type of ion enters the AC pores. The CV results of cations and anions indicated that capacitance originated from the contribution of the counter-ions or from the competition between counter-ions and co-ions, which were determined by the polarization of AC electrodes. An extensive understanding of the detailed charge storage mechanism of AC in BMIM-NTf₂ system was achieved by further combining the relative results of $BMIM^+$ and $NTf_2^-$ with the data obtained from the EQCM experiment. The perm-selective behavior alone was observed at the highly polarized potential regions (below −0.1 V/ref and above 0.9 V/ref), whereas perm-selective and ion swapping behaviors were both observed at the low polarized potential regions (within the intermediate potential window of −0.1 to 0.9 V/ref), as determined by the competition between cations and anions. This work provided an effective way to respectively characterize the charging behavior of cations and anions of ILs in ACs. We can imagine the characterization of more types of ions can be achieved by utilizing the corresponding SiO₂-grafted ILs. Our work is of great significance to the rational choice of IL electrolytes and efficient construction of high-performance EDLCs.

## Methods

**Preparation of silica nanoparticle-grafted ionic liquids**. SiO₂-MIM-Cl (**5**). 1-methylimidazole (**8**, 8.0 mL, 0.1 mmol) was added in a solution of (3-chloropropyl) trimethoxysilane (**7**, 18.2 mL, 0.1 mmol) in dimethylformamide (DMF, 120 mL) and the mixture was stirred at 80 °C for 48 h. After the reaction was completed, the solvent was evaporated under vacuum. The resulting viscous liquid was dissolved in H₂O and the organic byproducts were extracted with ether (Et₂O) and dichloromethane (CH₂Cl₂). The aqueous phase was evaporated under vacuum and the product **9** was obtained as yellow oil. Subsequently, an aqueous suspension of silica nanoparticles (200 mL, 2.5 wt%) was added in an aqueous solution of **9** (5 mL, 1.0 mol/L) slowly. The mixture was stirred at 80 °C for 12 h to afford SiO₂-MIM-Cl (**5**) as a white suspension without further purification.

SiO₂-NTf-Li (**6**). The trifluoromethanesulfonic anhydride (Tf₂O, 16.9 mL, 0.1 mmol) was slowly added in a solution of 3-(trimethoxysilyl)propan-1-amine (**12**, 17.5 mL, 0.1 mmol) and triethylamine (Et₃N, 28 mL, 0.2 mmol) in dried CH₂Cl₂ (150 mL) at −78 °C under dry argon atmosphere. The mixture was stirred for 1 h and then heated to room temperature. An aqueous of potassium hydroxide (KOH, 17.1 g, 0.3 mmol) was then added in this mixture and the resulting organic byproducts were extracted with CH₂Cl₂. The aqueous phase was neutralized with hydrochloric acid (HCl, 80 mL, 4 mol/L) and extracted with ethyl acetate. The organic phase was dried by magnesium sulfate (MgSO₄) and then concentrated to

afford a colorless oil. This oil was dissolved in anhydrous acetonitrile ($CH_3CN$, 150 mL) and lithium hydride (LiH, 1.2 g, 0.15 mmol) was then added in the solution slowly at 0 °C. The resulting solid byproducts and excess LiH were removed by filtration. After the solvent was evaporated under vacuum, the product **13** was obtained as white solid. Subsequently, an aqueous suspension of silica nanoparticles (200 mL, 2.5 wt%) was added in a solution of **13** (15 mL, 1.0 mol/L in $H_2O$) slowly. The mixture was stirred at 80 °C for 12 h to afford $SiO_2$-MIM-Cl (**6**) as a white suspension without further purification.

$SiO_2$-IL-$NTf_2$ (**1**). The suspension of $SiO_2$-MIM-Cl (**5**) was added in an aqueous solution of $LiNTf_2$ (**10**, 1.05 equiv. to **9** in previous step). After fully mixing, the resulting precipitation was separated by centrifugation. The solid product was purified by washing with acetone/$H_2O$ (v/v = 3/2) and centrifugation. The final traces of $H_2O$ were removed by lyophilized under vacuum to afford the product $SiO_2$-IL-$NTf_2$ (**1**).

$SiO_2$-IL-$PF_6$ (**2**). The suspension of $SiO_2$-MIM-Cl (**5**) was added in an aqueous solution of $LiPF_6$ (**11**, 1.05 equiv. to **9** in previous step). After fully mixing, the precipitated solid was separated by centrifugation. The solid product was purified by washing with acetone/$H_2O$ (v/v = 3/2) and centrifugation. The final traces of $H_2O$ were removed by lyophilized under vacuum to afford the product $SiO_2$-IL-$PF_6$ (**2**).

$SiO_2$-IL-BMIM (**3**). The suspension of $SiO_2$-$NTf_2$-Li (**6**) and $CH_2Cl_2$ (2 mL) were added in an aqueous solution of BMIMCl (**14**, 1.05 equiv. to **13** in previous step). After fully mixing, the precipitated solid was separated by centrifugation. The solid product was purified by washing with $H_2O$ and centrifugation. The final traces of $H_2O$ were removed by lyophilized under vacuum to afford the product $SiO_2$-IL-BMIM (**3**).

$SiO_2$-IL-$NBu_4$ (**4**). The suspension of $SiO_2$-$NTf_2$-Li (**6**) was added an aqueous solution of $NBu_4Cl$ (**15**, 1.05 equiv. to **13** in previous step). After fully mixing, the precipitated solid was separated by centrifugation. The product was purified by washing with acetone/petroleum ether (v/v = 2/3) and centrifugation, then washing again with $H_2O$ and centrifugation. The final traces of $H_2O$ were removed by lyophilized under vacuum to afford the product $SiO_2$-IL-$NBu_4$ (**4**).

**Material characterization**. $^1H$ NMR and $^{19}F$ NMR spectra were recorded in d-DMSO solution on a Bruker 400 MHz spectrometer. The spectral data were reported in ppm and calibrated by using DMSO (2.50 ppm) as internal reference for $^1H$ NMR. The nitrogen adsorption-desorption isotherm was measured on an ASAP 2020 M porosimeter (Micromeritics) at 77 K. The specific surface area (SSA) was calculated using the Brunauer–Emmett–Teller (BET) method and the pore size distribution (PSD) was calculated from the isotherm using the Barrett–Joyner–Halenda (BJH) method. Transmission electron microscopy (TEM) was conducted using a JEOL 2100 FEG microscope at 200 keV. Scanning electron microscopy (SEM) was conducted using a JEOL JSM 6701 F microscope. Energy dispersive X-ray spectroscopy (EDS) was conducted using a JSM-5601LV microscope. Inductively coupled plasma optical emission spectroscopy (ICP-OES) was conducted using a Leeman Prodigy7 spectrometer. Fourier transform infrared (FTIR) spectra were recorded using a Nexus 870 FTIR spectrometer. Thermogravimetric analysis (TGA) was carried out using a STA 449 C thermal analyzer at a heating rate of 10 °C min$^{-1}$ in air atmosphere. The electrical conductivity of IL electrolytes was measured by using a Mettler Toledo FE30 FiveEasy conductivity meter.

**Electrochemical measurements**. Electrochemical CV tests were carried out using an electrochemical workstation (CHI440C, Shanghai, China) in a closed three-electrode system, assembled in a glovebox under argon, with a silver disk as the quasi-reference electrode, a platinum foil (1 × 1 cm²) as the counter electrode and a carbon electrode as the working electrode. The carbon electrodes were prepared from YP-50F and SADC: 95 wt% carbon powder (2.0 mg), 5 wt% poly(tetra-fluoroethylene) (PTFE) and few drops of ethanol were homogeneously mixed. After allowing the solvent to evaporate, the resulting paste was pressed onto the foamed nickel current collector with an area of 1 cm² at 10 MPa. These carbon electrodes were dried at 60 °C for 12 h under vacuum. Before recording the data, each CV test was firstly cycled three times to make sure the system reached a steady sate.

*Capacitance calculations*. The specific capacitance $C$ (F g$^{-1}$) of CV curves was calculated by integration of current with respect to time, i.e., equation (1):

$$C = \frac{\int_0^{V/s} j\, \mathrm{d}t}{V} \qquad (1)$$

where $j$ is the gravimetric current density (A g$^{-1}$), $s$ is the scan rate (V s$^{-1}$), and $V$ is the potential window (V).

For EQCM measurements, a quartz crystal microbalance (Princeton, QCM922A) system combined with an electrochemical workstation (CHI440C, Shanghai, China) was used for simultaneous EQCM and CV measurements. These electrochemical tests were carried out using a carbon-coated quartz as the working electrode, a platinum foil (1 × 1 cm²) as the counter electrode, a silver disk as the quasi-reference electrode, and a solution of 0.25 M BMIM-$NTf_2$ in PC as the electrolyte. The carbon-coated quartz electrode was prepared from YP-50F: 90 wt% carbon powder, 10 wt% polyvinylidene fluoride (PVDF) in N-methyl-2-

pyrrolidone were homogeneously mixed. This slurry was dripped on an AT-cut gold-coated quartz crystal with a fundamental frequency of 9.00 MHz (the mass loading was 25–50 μg cm$^{-2}$), which was then dried at 60 °C for 12 h under vacuum (details see Supplementary Information). The corresponding CV tests were carried out at a scan rate of 5 mV s$^{-1}$ with simultaneous recording of the quartz resonance frequency. Before recording the data, each CV test was firstly cycled three times to make sure the system reached a steady sate.

**Data availability**. All relevant data supporting the findings of this study are available from the authors on request.

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

## Acknowledgements

This work was supported by the National Nature Science Foundations of China (21573265 and 21673263).

## Author contributions

X.B.Y. designed this work; Q.Y.D. carried out the organic synthesis and electrochemical experiments; B.J.Y. and J.W.L. made sodium alginate-derived AC; X.B.Y. and Q.Y.D. wrote the paper and analyzed the results. All authors contributed to the discussion of the results.

## Additional information

**Competing interests:** The authors declare no competing financial interests.

