## [Peer Review File · Nature Communications]

Reviewers' comments:

Reviewer #1 (Remarks to the Author):

The authors have grafted cations and anions of ionic liquids to SiO₂ nanoparticles to probe the transport of cations and anions in nano pores of carbon as this relates to charging behavior in supercapacitors. The authors suggest that many elegant new in site and in operando techniques (e.g., IR, NMR) are limited and therefore their results will provide more insight into the charging behavior in supercapacitors.

It is not clear that the other techniques are as limiting as the authors propose and it is not clear that the results presented (specifically anion and cation transport mechanisms in carbon nano pores) are new in comparison to what has already been published. Further more, the results from other techniques that have been published have been on capacitors (in operando) and in pure ionic liquid electrolyte. In this study, the authors do not investigate a fully functioning capacitor (in operando) and also do not examine pure ionic liquid electrolyte (diluted in propylene carbonate). Additionally, one may not be able to directly compare the movement of an ionic liquid to an ionic liquid grafted to a nanoparticle. There may be differences in these two different molecules.

Overall, the authors have introduced to many factors in this study they may not translate into useful fundamental results for an actual capacitor. Additionally, the authors are underestimating the impact of previous work in this field. Also, the style of writing does not appear to match that for a Nature Communications publication.

I would not recommend this for publication in Nature Communications.

Reviewer #2

Reviewer's report on the MS paper by Qingyun Dou et al "Silica-grafted ionic liquids: towards revealing the respective charging behaviors of cations and anions in supercapacitor".

EQCM technique possesses extraordinary high sensitivity in real time monitoring of compositional changes in charged nanoporous carbon electrodes. The contribution of counter- and co-ions into charge compensation mechanism in carbon nanopores is assessed dynamically, close to the conditions of operating supercapacitor cells. The conventional way to modify the dynamics, and hence to deeper understand the nature of ionic part of the electric double layer of carbon electrodes is the variation of the pore width-ion size ratio and also the electrode charge density.

The system with the single type of mobile ions in the solution described in the reviewed manuscript is extremely important for a better understanding of the fine mechanisms of nanoporous carbon charging. Three types of CV responses demonstrated in Fig. 4 convincingly demonstrative the deconvolution of the total CV response due to the contribution of two mobile ions (counter- and co-ions) into two individual CVs due to the participation of single mobile ions in the charging process. In my opinion, this beautiful result definitely deserves publication in such a high-impact factor journal as Nature Communications.

However, direct supporting evidence as to the nature of the ions dominating in the charge compensation should come from the EQCM study. Unfortunately, the EQCM analysis used by the authors is a bit under-developed and partially confused – see my comments below. My final decision about the suitability of this manuscript for publication in Nature Communications depends on the authors' ability to find satisfactory solutions to the problems listed below. At this stage, I recommend the major revision of the paper.

The most important comment: the transform from the frequency change to the related mass density change is not always (unconditionally) correct, it requires a rigorous substantiation (for a detailed discussion of the use of Sauerbrey's equation in the electrochemistry of porous energy-storage electrodes - see a recent review in *Electrochim. Acta*, Volume 232, 1 April 2017, Pages 271–284). I fully realize that the authors may not have access to the most powerful (and expensive) multi-harmonic acoustic analytical instrument, QCM-D (QCM with dissipation monitoring), however the instrument they used, i.e. Princeton, QCM922A, operating on the fundamental, ensures the measurement of the motional resistance (called in the users manual "resonance resistance"). The authors must present the proof that the change in the resonance resistance when charging their carbon electrodes from PZC to the vertex anodic and cathodic potentials (transforming these changes into the related resonance width change in Hz) remains much smaller than the simultaneous change of the related resonance frequency. This is the necessary condition for a purely inertial load described by the Sauerbrey equation. In case the authors will fail to present such a proof, I'll recommend submitting the manuscript to one of the more specialized technical journals.

Further important comments:

1. I am really surprised to see in the top panel of Fig. 6c repetition of panel Fig. 6a and in the bottom panel CV taken from other figures. I would expect to see at this very place, the analog of Fig. 6b related to the EQCM study of carbon charging in the presence of SiO₂-IL-single mobile anion or cation. The CV experiments in these systems have been done but not the related EQCM experiments (the latter were limited to the solutions with two mobile ions). The question is what is the aggregate state of SiO₂-IL-single mobile anion or cation system in PC? Is it a stable colloid system without adsorption on carbon electrode surface? In the case of the absence of the adsorption, such a colloid system would be sensed by QCM as having an increased viscosity.

2. An advanced EQCM analysis implies the use of the population change, $\Delta\Gamma$ (nmoles/cm²) as a function of the electrode charge rather than the mass change. Counter- and co-ions have different molecular masses which requires the use of two different Faradaic lines (like in Fig. 6 b). Since both counter and co-ions are single-charged ions, the Faradaic lines with the use of $\Delta\Gamma$ have the same slopes values but with the opposite sign. The authors cited the paper (ref. 37) in which this advanced approach has been well demonstrated.

3. The authors do not correctly cite the available literature. The role of co-ions in charge compensation mechanism in nanoporous carbons was proposed for the first time in *Chem. Phys. Chem* (2011), 12, 854-862 rather than in the subsequent papers. For the systems where desorption of co-ions plays the dominant role in charge compensation mechanism in nanoporous carbons, the symmetry between the population changes of cations and anions with respect to PZC is disturbed so that the minimum in $\Delta\Gamma$ or Δm shifts from the pzc to the potential of the so-called minimum mass changes. This effect described in detail in the above reference is clearly seen in Fig. 6b of the reviewed manuscript (pzc is shifted to the left with respect to this minimum).

4. The authors focus on the interplay between the role of counter- and co-ions using their original single-mobile ion design. However, in the previous studies similar role was proved for the systems with the different size of cations and ions (the authors correctly cite ref. 39 in this respect). Another example was previously reported in *Electrochem. Commun.*, (2010) 12, 1718-1721: the carbon surface bearing oxygen-containing functional groups can acquire opposite electric charge when pH of the solution is decreased. Despite the fact that the electrode potential shifts towards more negative values the dominant protonated groups (basically surface-localized species) promote the Cl⁻ anions desorption instead of the adsorption of cations.

5. Why the mass change curve in Fig. 6b has a flat minimum close to the potential of zero mass change rather than having an intersection of two straight Faradaic lines? Carbon is an EDLC electrode. Electric double layer models (Guy-Chapman-Stern model and its extension) were incorporated into the theory of ions population (mass) changes tracked by the most advanced EQCM-D instrument in a recent paper (J. Phys.: Condens. Matter (2016) 28, 114001. The authors can learn from this paper how differential charge efficiency is related to the slope of the ions mass changes curves – this will provide a proper explanation of the purely perm-selective behavior of carbon (“adsorption” model in the reviewed manuscript) observed at high electrode charge density whereas the perm-selectivity failure (“ion-exchange mechanism” in the reviewed paper) is rather typical for small electrode charge density. The term “ion-exchange” mechanism is a very unfortunate term, in reviewer’s opinion: ion’s swapping or perm-selectivity failure would be much better.

Minor comments:

1. I perfectly understand why the authors used PVdF binder in their composite electrode for EQCM experiments whereas for the CV measurements they used PTFE. However, the readers may not guess correctly the reason, it is thus better to explain the reason in the text of the manuscript.

2. The statement in lines 73, 74 “... EQCM only measure one specific parameter, the resonance frequency for EQCM” is not correct, motional resistance in your instrument or dissipation factor in EQCM-D is another channel of information in addition to the frequency changes - this is the background of non-gravimetric EQCM. In the context of the reviewed manuscript - see my “The most important comment” above.

3. Lines 233, 234: “...electrode in four ILs (BMIM-NTf2, BMIM-PF6, NBu4-NTf2 and NBu4-PF6) as shown in Figure 4a-d).” The last two compositions in PC are certainly not the ILs.

4. Lines 298, 299: “and the frequency response at PZC was set to zero.” See my note about the potential of zero mass change (Chem. Phys. Chem (2011), 12, 854-862).

5. Line 472: “...This slurry was dripped on an AT-cut gold-coated quartz crystal”. How the authors homogeneously distributed 5-8 microgram carbon coating? Can the authors show SEM image of this surface (I understood that SEM image in Fig. S1c in SI relates to carbon powder rather than to the coated electrode...).

6. Are the authors sure that they reached the complete impregnation of their porous carbon electrode with the solution?

7. EQCM always operates with averaged mass density change. For this reason the correct dimension of the sensitivity factor (line 184 in SI) should be ng/(Hz cm²).

Reviewer #3 (Remarks to the Author):

This paper provides an interesting means of separately monitoring the cations and anions in ionic liquid electrolytes used for characterizing the capacitive storage properties of activated carbon (AC). By grafting an anion or cation to a SiO₂ nanoparticle, the authors claim they are able to selectively store charge on activated carbon electrodes with the oppositely-charged, free ion. Through the use of cyclic voltammetry (CV) and EQCM, the authors propose that a combination of ion adsorption and ion exchange processes are involved in charge storage.

There is a curious omission in this paper in that the authors do not actually quantify their capacitance results. They show the experimental CV's, which is fine, but what is being compared here are data which are not normalized by sample weight. No specific capacitance values (F/g) are identified. Thus, when the authors show the differences between 'fitted' and 'tested' data, we do not know whether the differences involved are 2% or 50%. If the differences are 10% or less, then one must take into account sample variability, as keeping sample weights within 10% can be a challenge. Moreover, by not knowing the magnitude of the specific capacitance, we cannot establish whether the materials being tested by the authors are consistent with literature values or if they are significantly different. In the latter case, the authors would have to explain why their values for capacitive storage are different from what others have reported.

Another point that must be addressed is whether the grafted materials effectively lead to zero current. There is the assumption that the grafted ions do not contribute to charge storage, but this is not verified in the paper. It is essential that the authors perform such control experiments in order to give validity to their central hypothesis regarding the selectivity of charge storage.

Response to referees

Response to reviewer 1:

Reviewer's Comments: "The authors have grafted cations and anions of ionic liquids to SiO₂ nanoparticles to probe the transport of cations and anions in nano pores of carbon as this relates to charging behavior in supercapacitors. The authors suggest that many elegant new in site and in operando techniques (e.g., IR, NMR) are limited and therefore their results will provide more insight into the charging behavior in supercapacitors.

(a) It is not clear that the other techniques are as limiting as the authors propose and it is not clear that the results presented (specifically anion and cation transport mechanisms in carbon nano pores) are new in comparison to what has already been published. (b) Furthermore, the results from other techniques that have been published have been on capacitors (in operando) and in pure ionic liquid electrolyte. In this study, the authors do not investigate a fully functioning capacitor (in operando) and also do not examine pure ionic liquid electrolyte (diluted in propylene carbonate). (c) Additionally, one may not be able to directly compare the movement of an ionic liquid to an ionic liquid grafted to a nanoparticle. There may be differences in these two different molecules. (d) Overall, the authors have introduced to many factors in this study they may not translate into useful fundamental results for an actual capacitor. (e) Additionally, the authors are underestimating the impact of previous work in this field. (f) Also, the style of writing does not appear to match that for a Nature Communications publication.

I would not recommend this for publication in Nature Communications."

Reply: Thank you very much for your review. However, it is difficult for us to fully agree with you. The detailed answers are listed as follows.

(a) Undoubtedly, for the study on the charge storage mechanisms of supercapacitors, great contributions have been made by using different techniques such as computer simulation, NMR, EQCM, IR and SANS. The advantages and the respective

characteristics of these techniques have been also described in our initial manuscript. In addition, each technique actually has its certain analytical limitation, which had been well summarized by C. P. Gray et al. (*J. Am. Chem. Soc.* **138**, 5731-5744 (2016)), and we mentioned but not exaggerated these limitations in the introduction section of our initial manuscript. But we should apologize for that a few words or sentences were not appropriately used in our initial introduction, so that you felt that we are underestimating the impact of these previous works. These errors have been carefully modified in the revised manuscript:

1. *“However, these simulations need to build on some necessary simplifications and assumptions in advance, and they cannot accurately describe the complex porous/and electronic structures of ACs. Thus, the obtained results are too idealistic.”* was modified as *“But these simulations need to build on necessary simplifications to balance simulation time and accuracy, so they usually do not include all details of complex porous/and electronic structures of ACs at experimental conditions.”*
2. *“IR just can detect the bulk electrolyte surrounding AC particles and cannot directly detect the in-pore ions.”* was modified as *“IR cannot directly detect the in-pore ions.”*
3. *“...SANS and EQCM only measure one specific parameter, the resonance frequency for EQCM and the diffraction intensity for SANS, and this parameter depends on...”* was modified as *“However, in SANS and EQCM, the parameter, diffraction intensity and resonance frequency, respectively, depends on...”*
4. *“However, ex-situ NMR investigation needs to disassembly the EDLC cell in advance, which cannot avoid the problems of self-discharge and solvent evaporation, resulting in an obvious deviation existing in the subsequent NMR measurement.”* was modified as *“However, ex-situ NMR investigation needs to disassembly the EDLC cell in advance, which cannot avoid the problems of self-discharge and solvent evaporation, thus a probable deviation existing in the subsequent NMR measurement.”*

With regard to novelty, herein we provide a totally new strategy (silica nanoparticle-grafting ionic liquids (ILs)) to respectively monitor the characteristics of

cations and anions of ILs in carbon pores by allowing only one type of ions to enter the pores in the charging process. The system with the single type of mobile ions in the electrolyte is extremely important for a better understanding of the fine mechanisms of nanoporous carbon charging. In our paper, three types of CV responses demonstrated convincingly demonstrative the deconvolution of the total CV response due to the contribution of two mobile ions (counter- and co-ions) into two individual CVs due to the participation of single mobile ions in the charging process. In summary, this work provides a completely new and effective method for revealing the sorption mechanisms of cations and anions of ILs in EDLCs and will be useful to choose more suitable IL electrolytes for high-performance supercapacitor devices.

(b) It is true that the reported mechanism results from other techniques such as NMR, IR and SANS were operated on capacitors. However, the mechanism studies using EQCM technique, as far as we know, were carried out by characterizing the property of a single electrode (*J. Am. Chem. Soc.* **136**, 8722-8728 (2014); *J. Am. Chem. Soc.* **132**, 13220-13222 (2010); *J. Phys. Chem. C* **117**, 14876-14889 (2013)). These studies showed that the contribution of counter- and co-ions into charge compensation mechanism in carbon nanopores can be assessed dynamically, close to the conditions of operating supercapacitor cells. In addition, the study focusing on a single electrode is well suited for characterizing the intrinsic electrochemical property of the electrode in a given electrolyte (*Science* **350**, 1508-1513 (2015); *Nature Energy* **2**, 17105 (2017)), and such strategy is also important and instructive for the construction of the fully functioning capacitor.

With regard to electrolyte, on one hand, it is easy to find the cases that some experimental techniques did not operate in pure IL electrolyte. For example, using *in situ* NMR technique, C. P. Grey et al. have revealed the storage mechanism of supercapacitor in $\text{PEt}_4\text{-BF}_4/\text{ACN}$ electrolyte (*Nat. Mater.* **14**, 812-819 (2015)). EQCM technique was used by P. Simon et al. to characterize ion adsorption in both pure and

solvated EMIM-NTf₂ electrolyte (*J. Am. Chem. Soc.* **136**, 8722-8728 (2014)). SANS technique was also operated in a solvated TEA-BF₄/ACN electrolyte (*ACS Nano* **8**, 2495-2503 (2014)). On the other hand, we believe that the investigation of the storage mechanism of supercapacitors in ionic liquid-organic electrolytes is also important. Dilution of a given IL with an organic solvent can result in a lower viscosity and a higher conductivity of electrolyte, thus enable a better overall electrochemical performance of supercapacitor (*Angew. Chem. Int. Ed.*, **54**, 4806-4809 (2015); *Electrochim. Acta* **153**, 426-432 (2015)).

(c) We cannot accept your opinion that one may not be able to directly compare the movement of an ionic liquid to an ionic liquid grafted to a nanoparticle. In contrast, we believe that the comparison between grafted ionic liquids and conventional ionic liquids is reasonable. In our system, by allowing single ions to participate (enter in the pores of carbon) in the charging process, we have separately characterized two important properties of cations and anions, i.e. the capacitance contribution and the acting potential window. In our manuscript, the data of CV curves were showed in specific capacitance (F/g) after normalizing by sample weight. It was found that the differential capacitance of YP-50F electrode in both grafted ILs and conventional ILs was close to 100 F/g (Figure 3 and 4), confirming that the capacitance contributions of the free ions were not obviously influenced after grafting the other ions to SiO₂ nanoparticles. The difference between cations and anions was their acting potential windows, i.e., the cations and anions contributed to charge storage at negatively polarized potential windows and positively polarized potential windows, respectively. These obtained results were reasonable when considering the electrostatic repulsion of the electrode surface with co-ions (ions with the sign of their charge the same as that of the electrode surface). The slight difference of acting potential windows of ions with the same charge may be attributed to the different interactions between electrode surface and ions, such as Van der Waals force.

(d) As we mentioned before, an effective method to respectively characterize the

charging behaviors of cations and anions in ACs has been showed in this work. The obtained results are very important and useful for an in-depth understanding of the charge storage mechanism of supercapacitors and a better choice of suitable ionic liquid electrolyte for high-performance devices.

(e) As discussed in part (a), we greatly appreciate the contributions of previous works in this field and really apologize for the inappropriate description of other techniques in our initial manuscript. The inappropriate words and sentences have been carefully modified in the introduction of the revised manuscript.

(f) We should say sorry for that we did not fully understand the specific meaning of “the style of writing” you mentioned. We had carefully read the “author instructions” of Nature Communications before preparing this manuscript. It was then prepared according to the template shown in the “author instructions” of Nature Communications. However, we will be very grateful and willing to modify our manuscript if any detailed problem is pointed out.

Response to reviewer 2:

Reviewer’s summary remark: “Reviewer’s report on the MS paper by Qingyun Dou et al “Silica-grafted ionic liquids: towards revealing the respective charging behaviors of cations and anions in supercapacitor.

EQCM technique possesses extraordinary high sensitivity in real time monitoring of compositional changes in charged nanoporous carbon electrodes. The contribution of counter- and co-ions into charge compensation mechanism in carbon nanopores is assessed dynamically, close to the conditions of operating supercapacitor cells. The conventional way to modify the dynamics, and hence to deeper understand the nature of ionic part of the electric double layer of carbon electrodes is the variation of the pore width-ion size ratio and also the electrode charge density.

The system with the single type of mobile ions in the solution described in the

reviewed manuscript is extremely important for a better understanding of the fine mechanisms of nanoporous carbon charging. Three types of CV responses demonstrated in Fig. 4 convincingly demonstrate the deconvolution of the total CV response due to the contribution of two mobile ions (counter- and co-ions) into two individual CVs due to the participation of single mobile ions in the charging process. In my opinion, this beautiful result definitely deserves publication in such a high-impact factor journal as Nature Communications.

However, direct supporting evidence as to the nature of the ions dominating in the charge compensation should come from the EQCM study. Unfortunately, the EQCM analysis used by the authors is a bit under-developed and partially confused – see my comments below. My final decision about the suitability of this manuscript for publication in Nature Communications depends on the authors' ability to find satisfactory solutions to the problems listed below. At this stage, I recommend the major revision of the paper.”

Reply: We appreciate deeply for your careful review on our manuscript as well as the highly positive evaluation on our work. After reading your detailed comments, we know that you are the top-class scientist in EQCM field, and your professional suggestions are very helpful for us to reinforce our manuscript. According to your comments, the initial manuscript has been revised carefully. Detailed answers are listed as follows.

The most important comment: “the transform from the frequency change to the related mass density change is not always (unconditionally) correct, it requires a rigorous substantiation (for a detailed discussion of the use of Sauerbrey's equation in the electrochemistry of porous energy-storage electrodes - see a recent review in *Electrochim. Acta*, Volume 232, 1 April 2017, Pages 271–284). I fully realize that the authors may not have access to the most powerful (and expensive) multi-harmonic acoustic analytical instrument, QCM-D (QCM with dissipation monitoring), however the instrument they used, i.e. Princeton, QCM922A, operating on the fundamental, ensures the measurement of the motional resistance (called in the users manual

“resonance resistance”). The authors must present the proof that the change in the resonance resistance when charging their carbon electrodes from PZC to the vertex anodic and cathodic potentials (transforming these changes into the related resonance width change in Hz) remains much smaller than the simultaneous change of the related resonance frequency. This is the necessary condition for a purely inertial load described by the Sauerbrey equation. In case the authors will fail to present such a proof, I’ll recommend submitting the manuscript to one of the more specialized technical journals.”

Reply: Thank you very much for your important comment, and we are very sorry for not providing the important information of the resonance resistance change or the resonance width change. Figure R1a shows that, when the YP-50F-coated quartz was cycled between -0.2 V to 0.9 V, the resonance resistance change (ΔR) was about 6 Ω . The resonance resistance change (ΔR) can be transformed into the related resonance width change (ΔW) by equation (1):

$$\Delta W = \frac{16 A e_{26}^2 \rho_q f_q^3}{\pi Z_q^3} \cdot \Delta R \quad (1)$$

where A is the area of active surface (0.198 cm^2), e_{26} is the piezoelectric stress coefficient ($9.65 \times 10^{-2} \text{ C m}^{-2}$), ρ_q is the quartz crystal density (2.65 g cm^{-3}), f_q is the reference frequency (9.0 MHz), and Z_q is the acoustic wave impedance ($8.8 \times 10^6 \text{ kg m}^{-2} \text{ s}^{-1}$). Figure R1b shows that the resonance width change (ΔW), obtained by multiplying the resonance resistance (ΔR) by a factor of 14, is much smaller than the simultaneous resonance frequency change (Δf), indicating a gravimetric behavior of the quartz-crystal microbalance.

Figure R1 | The original data of resonance frequency and resonance resistance vs potential (**a**), and the related frequency change Δf and resonance width change ΔW vs potential (**b**) obtained for YP-50F-coated quartz electrode in BMIM-NTf₂/PC.

Further important comments:

1 “I am really surprised to see in the top panel of Fig. 6c repetition of panel Fig. 6a and in the bottom panel CV taken from other figures. I would expect to see at this very place, the analog of Fig. 6b related to the EQCM study of carbon charging in the presence of SiO₂-IL-single mobile anion or cation. The CV experiments in these systems have been done but not the related EQCM experiments (the latter were limited to the solutions with two mobile ions). The question is what is the aggregate state of SiO₂-IL-single mobile anion or cation system in PC? Is it a stable colloid system without adsorption on carbon electrode surface? In the case of the absence of the adsorption, such a colloid system would be sensed by QCM as having an increased viscosity.”

Reply: Thanks for your valuable comment. In our initial manuscript, Figure 6c showed the combination of EQCM results with the results obtained from the system with the single type of mobile ions for the purpose of an in-depth explanation of the charge storage mechanism of YP-50F electrode in BMIM-NTf₂/PC electrolyte. However, inspired by your comments, we found that the previous exhibition was more or less confused, and in our revised manuscript this section was carefully modified (Figure 7 in the revised manuscript).

The state of SiO₂-IL-single mobile ions system in PC could be regarded as a uniform dispersion of charged silica nanoparticles in PC solvent (*J. Mater. Chem.* **22**, 4066-4072 (2012)). In our system, the silica nanoparticles-grafted ILs could be dispersed in PC with a dilute concentration of 0.25 M (Figure R2). However, the charged silica nanoparticles could be also adsorbed on the apparent electrode surface (note that they cannot access the internal surface of carbon pores) and contribute very low and negligible specific capacitance during the CV tests (Figure R3). An increased viscosity near the electrode surface may also occur over several CV cycles. More

importantly, the adsorption of the charged silica nanoparticles (silica particles have much higher mass than molecules of ionic liquids) could lead to an unusual mass increase of carbon electrode, thus the data obtained by EQCM experiment would be useless because the use of EQCM for the study of EDLC mechanism under the assumption that the mass change of carbon electrode is linked exclusively to the ions adsorption. Based on the above considerations, we did not employ the EQCM study of carbon charging in the presence of SiO₂-IL-single mobile ions.

Figure R2 | The photographs of (a) the fluid-like silica nanoparticles-grafted IL/PC electrolyte and (b) the Tyndall scattering effect via a red laser beam, indicating the uniform dispersion of charged silica nanoparticles in PC.

Figure R3 | Comparison of the CV curves of YP-50F electrode in SiO₂-MIM-SiO₂-NTf/PC and in BMIM-NTf₂/PC electrolytes within the operating potential window of -0.8~1.2 V with the same scan rate of 5 mV s⁻¹. SiO₂-MIM-SiO₂-NTf is a synthesized materials in which both cations and anions were grafted to SiO₂ (The structure of SiO₂-MIM-SiO₂-NTf was confirmed by ¹H NMR (Figure S4)). It was found that the charged silica nanoparticles could also

contribute very and negligible specific capacitance, indicating they could be also adsorbed on the apparent electrode surface

Figure R4 | The ^1H NMR spectrum of $\text{SiO}_2\text{-MIM-SiO}_2\text{-NTf}$.

2. “An advanced EQCM analysis implies the use of the population change, $\Delta\Gamma$ (nmoles/ cm_2) as a function of the electrode charge rather than the mass change. Counter-and co-ions have different molecular masses which requires the use of two different Faradaic lines (like in Fig. 6 b). Since both counter and co-ions are single-charged ions, the Faradaic lines with the use of $\Delta\Gamma$ have the same slopes values but with the opposite sign. The authors cited the paper (ref. 37) in which this advanced approach has been well demonstrated.”

Reply: Thanks for your valuable and constructive suggestion. In our revised manuscript, the population change, $\Delta\Gamma$ (nmoles/ cm_2) was used according to your comment, and we found that the EQCM analysis became more concise.

3. “The authors do not correctly cite the available literature. The role of co-ions in charge compensation mechanism in nanoporous carbons was proposed for the first time in Chem. Phys. Chem (2011), 12, 854-862 rather than in the subsequent papers.

For the systems where desorption of co-ions plays the dominant role in charge compensation mechanism in nanoporous carbons, the symmetry between the population changes of cations and anions with respect to PZC is disturbed so that the minimum in $\Delta\Gamma$ or Δm shifts from the pzc to the potential of the so-called minimum mass changes. This effect described in detail in the above reference is clearly seen in Fig. 6b of the reviewed manuscript (pzc is shifted to the left with respect to this minimum).”

Reply: Thanks for your comment and we are very sorry for not correctly citing the literature (*Chem. Phys. Chem.*, **12**, 854-862 (2011)). This literature discovered the dominant role of anion desorption when bulky cations were used, which is very important for us to explain the phenomenon in our work (pzc is shifted to the left with respect to pzmc). Thus, we cited this literature in our revised manuscript and made corresponding explanation

4. “The authors focus on the interplay between the role of counter- and co-ions using their original single-mobile ion design. However, in the previous studies similar role was proved for the systems with the different size of cations and ions (the authors correctly cite ref. 39 in this respect). Another example was previously reported in *Electrochem. Commun.*, (2010) **12**, 1718-1721: the carbon surface bearing oxygen-containing functional groups can acquire opposite electric charge when pH of the solution is decreased. Despite the fact that the electrode potential shifts towards more negative values the dominant protonated groups (basically surface-localized species) promote the Cl⁻ anions desorption instead of the adsorption of cations.”

Reply: Thanks for your valuable suggestion. The literature (*Electrochem. Commun.* **12**, 1718-1721 (2010)) demonstrated that Cl⁻ anion solely contributes to charge compensation mechanism in mixed CsCl+HCl solutions, and the results is similar to our system with single-mobile ion. Thanks again for providing this important literature. We also cited this literature and added the relative demonstration in our revised manuscript.

5. “Why the mass change curve in Fig. 6b has a flat minimum close to the potential of zero mass change rather than having an intersection of two straight Faradaic lines? Carbon is an EDLC electrode. Electric double layer models (Guy-Chapman-Stern model and its extension) were incorporated into the theory of ions population (mass) changes tracked by the most advanced EQCM-D instrument in a recent paper (J. Phys.: Condens. Matter (2016) 28, 114001. The authors can learn from this paper how differential charge efficiency is related to the slope of the ions mass changes curves – this will provide a proper explanation of the purely perm-selective behavior of carbon (“adsorption” model in the reviewed manuscript) observed at high electrode charge density whereas the perm-selectivity failure (“ion-exchange mechanism” in the reviewed paper) is rather typical for small electrode charge density. The term “ion-exchange” mechanism is a very unfortunate term, in reviewer’s opinion: ion’s swapping or perm-selectivity failure would be much better.”

Reply: We appreciate deeply for your kind and professional explanation on the charge compensation mechanism of EDLC. Your suggestions are very important to reinforce our manuscript. The flat minimum close to pzmc in Figure 6c implies the involvement of both co-ions (BMIM^+) desorption and counter-ions (NTf_2^-) adsorption in the charge storage mechanism. Moreover, after learning from the literature you suggested, we have modified our initial explanations in the revised manuscript (please see the detailed modifications in the text which have been marked by yellow color). Among them, the term “ion-exchange” mechanism was carefully modified as well.

Minor comments:

1. “I perfectly understand why the authors used PVdF binder in their composite electrode for EQCM experiments whereas for the CV measurements they used PTFE. However, the readers may not guess correctly the reason, it is thus better to explain the reason in the text of the manuscript.”

Reply: Thanks for your valuable and constructive suggestion. In our revised manuscript (see Supplementary Information), the reason for using different binder was added.

On the one hand, PTFE binder is generally used for the preparation of carbon electrodes of supercapacitors (*J. Am. Chem. Soc.* **137**, 7231-7242 (2015); *ChemSusChem* **7**, 3053-3062 (2014)), and the use of PTFE binder was beneficial to accurate control of the mass loading of the carbon electrodes, which is suitable for the comparison of CV curves in different electrolytes. On the other hand, using PTFE binder makes it difficult to homogeneously distribute carbon powder on Au-coated quartz crystal surface because PTFE is generally dissolved in water but the Au-coated quartz crystal surface is hydrophobic. Thus, we used PVdF binder dissolved in organic solvent (N-methyl-2-pyrrolidone) for EQCM experiments. Moreover, it has been demonstrated that a rigid electrode can be obtained when the carbon powder mixed with PVdF binder, which is the necessary condition for gravimetric application of EQCM (*J. Phys. Chem. C* **117**, 14876-14889 (2013); *Electrochim. Acta.* **232**, 271–284 (2017)).

2. “The statement in lines 73, 74 “... EQCM only measure one specific parameter, the resonance frequency for EQCM” is not correct, motional resistance in your instrument or dissipation factor in EQCM-D is another channel of information in addition to the frequency changes - this is the background of non-gravimetric EQCM. In the context of the reviewed manuscript - see my “The most important comment” above.”

Reply: Thanks for your valuable comment and we should say sorry for our incorrect description of EQCM technique. In our revised manuscript, the sentence “*However, SANS and EQCM only measure one specific parameter, the resonance frequency for EQCM and the diffraction intensity for SANS, and this parameter depends on...*” was modified as “*However, in SANS and EQCM, the parameter, diffraction intensity and resonance frequency, respectively, depends on...*”

3. “Lines 233, 234: “...electrode in four ILs (BMIM-NTf₂, BMIM-PF₆, NBu₄-NTf₂ and NBu₄-PF₆) as shown in Figure 4a-d).” The last two compositions in PC are certainly not the ILs.”

Reply: Thanks for your comment and we are sorry for our incorrect description. In

our revised manuscript, "...electrode in four ILs (BMIM-NTf₂, BMIM-PF₆, NBu₄-NTf₂ and NBu₄-PF₆) as shown in Figure 4a-d)." was modified as "...electrode in four organic electrolytes (BMIM-NTf₂, BMIM-PF₆, NBu₄-NTf₂ and NBu₄-PF₆ in PC) as shown in Figure 4a-d)." Moreover, similar errors in our initial manuscript were also carefully corrected.

4. "Lines 298, 299: "and the frequency response at PZC was set to zero." See my note about the potential of zero mass change (Chem. Phys. Chem (2011), 12, 854-862)."

Reply: Thanks for your valuable and constructive comment. In our revised manuscript, we changed the initial sentence. the frequency response at the potential of zero mass change (pzmc), i.e. the potential corresponding to the minimum mass change, was set to zero.

5. "Line 472: "...This slurry was dripped on an AT-cut gold-coated quartz crystal". How the authors homogeneously distributed 5-8 microgram carbon coating? Can the authors show SEM image of this surface (I understood that SEM image in Fig. S1c in SI relates to carbon powder rather than to the coated electrode...)"

Reply: Thanks for your constructive suggestion. With regard to the preparation of the carbon-coated quartz electrode, it is important to make sure that the materials (i.e. carbon powder, PVdF and N-methyl-2-pyrrolidone (DMP) solvent) were strictly dried and the operation was carried out in a dry environment (in the oven, for example). It is because the presence of water could cause the aggregation of the binder, which is unfavorable to obtain a uniform coating. In our experiment, the slurry prepared by mixing 90 wt% carbon powder, 10 wt% PVdF in DMP and drops of DMP (the total volume of DMP was about 80 μ L with respect to 10 mg carbon powder) was manually coated on the quartz crystal. Figure R5 shows the SEM images of YP-50F carbon coated on the quartz crystal surface.

Figure R5 | The SEM images of YP-50F on the quartz crystal surface.

6. “Are the authors sure that they reached the complete impregnation of their porous carbon electrode with the solution?”

Reply: Thanks for your valuable question. As described in our initial manuscript, each CV test (besides the EQCM experiments) was firstly cycled three times and a steady state was then obtained before recording the data. Thus we think that the porous carbon electrode was completely impregnated with the solution.

7. “EQCM always operates with averaged mass density change. For this reason the correct dimension of the sensitivity factor (line 184 in SI) should be $\text{ng}/(\text{Hz cm}^2)$ ”

Reply: Thanks for your valuable suggestion. For the QCM922’s 9MHz crystals, the sensitivity factor is $5.682 \text{ ng}/\text{Hz cm}^2$, which was used in our revised manuscript.

Response to reviewer 3:

Reviewer’s summary remark: “This paper provides an interesting means of separately monitoring the cations and anions in ionic liquid electrolytes used for characterizing the capacitive storage properties of activated carbon (AC). By grafting an anion or cation to a SiO_2 nanoparticle, the authors claim they are able to selectively store charge on activated carbon electrodes with the oppositely-charged, free ion. Through the use of cyclic voltammetry (CV) and EQCM, the authors propose that a combination of ion adsorption and ion exchange processes are involved in charge storage.”

Reply: We appreciate deeply for your kind and careful review on our manuscript as well as the positive evaluation on our work. Your comments and suggestions are very helpful to reinforce our manuscript. We have considered your comments carefully and the initial manuscript has been revised according to your comments. The detailed answers are listed as follows.

Comment 1: “There is a curious omission in this paper in that the authors do not actually quantify their capacitance results. They show the experimental CV’s, which is fine, but what is being compared here are data which are not normalized by sample weight. No specific capacitance values (F/g) are identified. Thus, when the authors show the differences between ‘fitted’ and ‘tested’ data, we do not know whether the differences involved are 2% or 50%. If the differences are 10% or less, then one must take into account sample variability, as keeping sample weights within 10% can be a challenge. Moreover, by not knowing the magnitude of the specific capacitance, we cannot establish whether the materials being tested by the authors are consistent with literature values or if they are significantly different. In the latter case, the authors would have to explain why their values for capacitive storage are different from what others have reported.”

Reply: Thank you very much for your kind and constructive suggestions. In our revised manuscript, the data of CV curves were normalized by sample weight. It should be mentioned that, in our experiments (except for EQCM experiment), all carbon electrodes were prepared with the same carbon mass loading, thus the current values were proportional to the specific capacitance values. So the obtained CV results in our initial manuscript were not changed after normalizing the CV data. Based on the normalized CV curves, only counter-ions contributed capacitance at the high polarized regions where the ‘fitted’ and ‘tested’ data were almost the same (the small difference between them on the reversal of potential may be attributed to the decreased ionic conductivity of SiO₂-grafted ILs), and cations and anions competed at low polarized regions where the ‘fitted’ data were significantly larger than the ‘tested’ data (the maximum values of the ‘fitted’ data were 1.7-2 times higher than the

corresponding values of the ‘tested’ data, Figure 4 in the revised manuscript).

The specific capacitance values of YP-50F electrode in BMIM-NTf₂/PC and BMIM-PF₆/PC electrolytes were about 95 F/g (calculated based on the CV curves), which were consistent with literature values (*Nat Mater.* **14**, 812-819, (2015); *J. Am. Chem. Soc.* **137**, 7231-7242 (2015)). While smaller specific capacitance values of 85 F/g were obtained in NBu₄-NTf₂/PC and NBu₄-PF₆/PC electrolytes, which could be attributed to the larger ion size of NBu₄⁺, thus smaller capacitance contribution of NBu₄⁺ when compared with that of BMIM⁺ (Supplementary, Figure S9a).

Comment 2: “Another point that must be addressed is whether the grafted materials effectively lead to zero current. There is the assumption that the grafted ions do not contribute to charge storage, but this is not verified in the paper. It is essential that the authors perform such control experiments in order to give validity to their central hypothesis regarding the selectivity of charge storage.”

Reply: Thank you very much for your valuable and constructive suggestion. Inspired by your comment, we synthesized a material named as SiO₂-MIM-SiO₂-NTf in which both cations and anions were grafted to SiO₂, and its chemical structure was confirmed by ¹H NMR spectrum (Figure R4). The SiO₂-MIM-SiO₂-NTf can only be dissolved in PC solvent with a low solubility. Although it was hard to accurately determine its concentration in PC because this grafted material may have different solubility in PC and *d*-DMSO (*d*-DMSO was used as the solvent for ¹H NMR experiment), a very dilute SiO₂-MIM-SiO₂-NTf/PC electrolyte was still obtained. The CV curve of YP-50F electrode in this SiO₂-MIM-SiO₂-NTf/PC electrolyte exhibited a distorted rectangular shape which may be due to the slow mobility of the grafted ions. Comparison of this CV curve with the CV curve in BMIM-NTf₂/PC electrolyte showed that the grafted ions led to a very low and negligible specific capacitance (or current) (Figure R3). Here, it should be mentioned that the grafted ions led to significantly decreased but not zero specific capacitance because the grafted ions could be also adsorbed on the apparent electrode surface although they cannot access the internal surface of carbon pores.

The above results were in accordance with the CV curves of YP-50F electrode in four SiO₂-grafted ILs (Figure 3 in the revised manuscript). We found that the capacitance contributed by cations (BMIM⁺, NBU₄⁺ Figure 3a, 3c) and anions (NTf₂⁻, PF₆⁻ Figure 3b, 3d) at high positively polarized potential and negatively polarized potential, respectively, was significantly decreased. And we defined the potential at which the maximum slope took place as the upper acting potential and the lower acting potential for cations and anions, respectively. As described in our initial manuscript, the similar results reported by Levi et al. (*J. Am. Chem. Soc.* **132**, 13220-13222 (2010)) showed that the charge at the negatively polarized electrode also rapidly decreased because of the inaccessibility of bulky TOA⁺ cations to most of the carbon pores, confirming our hypothesis that approximately only one ions contributed to the charge storage mechanism.

Figure R3 | Comparison of the CV curves of YP-50F electrode in SiO₂-MIM-SiO₂-NTf/PC and in BMIM-NTf₂/PC electrolytes within the operating potential window of -0.8~1.2 V with the same scan rate of 5 mV s⁻¹.

Figure R4 | The ^1H NMR spectrum of $\text{SiO}_2\text{-MIM-SiO}_2\text{-NTf}$.

REVIEWERS' COMMENTS:

Reviewer #1 (Remarks to the Author):

Many of the issues regarding the lack of novelty were not addressed in the authors' responses. Many of their responses stated that they disagree with the reviewer and that they "believe" that it is novel. This paper is not at the same level as the other papers published in JACS regarding new novel techniques to measure ionic liquids in fully functioning capacitors both in situ and in operando. I cannot recommend this publication in Nature Communications.

Reviewer #2 (Remarks to the Author):

The authors provided full and comprehensive answers to my questions. Among them the most important is proving the validity of the gravimetric approach (i.e. the use of the Sauerbrey's equation). The manuscript presents a significant contribution to the material science of energy storage devices. I have a pleasure to recommend the paper for the publication in Nature Communications.

Reviewer #3 (Remarks to the Author):

The authors have done a conscientious job of responding to my initial review. While the results provide better insight regarding the charging behavior, it is also likely that this approach will not lead to improvements in charge storage.

Reviewer #4 (Remarks to the Author):

The manuscript by Dou et al. presents an interesting novel way to separate the contributions of anions and cations in the charging mechanism of supercapacitor electrodes. The key advance here is using ionic solutions where one type of ion is specifically designed to be too large to fit inside the micropores of the electrode. As the authors acknowledge (ref 39), studies have been carried out previously on electrolytes with large ion size differences and similar observations have been reported. However, this work represents (to my knowledge) the first instance where this phenomenon has been exploited in a systematic way to study the charging mechanism. Experimental methods for gaining such information about the behavior of ions in AC electrodes during charging are still not common and this work has provided some important fundamental insight, and is an important contribution to the supercapacitor field. I therefore believe it should be published in Nature Communications subject to addressing a few comments:

- There are several mistakes or use of unclear English / grammar throughout and I recommend that this paper is proof-read by a native speaker prior to publication.
- In particular, I find the description of the "assembled" CV curves and "fitting" curves quite unclear and it took me a few re-reads to understand what they meant. I would suggest referring to "summed" CV curves or similar.
- The authors state "In addition, it can be found that the smaller capacitance of the fitting curves than those of the conventional electrolytes when the direction of the scan rate is reversed." I cannot see what this is referring to, and the term "direction of scan rate" is scientifically meaningless. This part needs to be explained better.

- I agree with reviewer 1's point about the authors' description of the limitations of other techniques and I still feel that the introduction dwells more than necessary on the drawbacks of other methods which have contributed greatly to the current understanding. The method proposed in the current work also has its drawbacks - it seems rather laborious and complicated, and does not provide molecular-level detail of other spectroscopic techniques. I think a better way to frame the current work would be to focus on the complementary and additional knowledge it brings to the other methods that have also provided great insight.

- I do not agree with reviewer 1's criticism of the fact that the work does not examine a pure IL. While it is true that ILs are of great interest as supercapacitor electrolytes, solvated electrolytes are more widely used and remain poorly understood. I do not see that it is "better" to study ILs. The main aim of the manuscript (and of much of the wider supercapacitor community) is simply to gain a better understanding of the behavior of ions at electrochemical interfaces, which still remains a great challenge. In some ways solvated electrolytes are even more complex than ILs with solvation/desolvation phenomena potentially contributing to the energy storage process, and so I do not see this as a negative aspect of the work.

Response to referees

Response to reviewer 1:

Reviewer's Comment: "Many of the issues regarding the lack of novelty were not addressed in the authors' responses. Many of their responses stated that they disagree with the reviewer and that they "believe" that it is novel. This paper is not at the same level as the other papers published in JACS regarding new novel techniques to measure ionic liquids in fully functioning capacitors both in situ and in operando. I cannot recommend this publication in Nature Communications."

Reply: Thank you again for your review. As we mentioned before, in our manuscript, we report a way to separately monitor cations or anions by allowing only one type of ions to enter carbon pores. We believe that the characterization of the respective charging behaviors of ions will provide a better insight into the storage mechanism of supercapacitors.

Response to reviewer 2:

Reviewer's Comment: "The authors provided full and comprehensive answers to my questions. Among them the most important is proving the validity of the gravimetric approach (i.e. the use of the Sauerbrey's equation). The manuscript presents a significant contribution to the material science of energy storage devices. I have a pleasure to recommend the paper for the publication in Nature Communications."

Reply: Thank you very much for your review as well as the positive evaluation on our work. Also, we appreciate deeply for your previous professional and constructive suggestions in which are all very useful to reinforce our manuscript. Many thanks again.

Response to reviewer 3:

Reviewer's Comment: "The authors have done a conscientious job of responding to my initial review. While the results provide better insight regarding the charging behavior, it is also likely that this approach will not lead to improvements in charge

storage.”

Reply: Thank you very much for your review and the positive evaluation on our work. In this paper, the strategy of designing SiO₂-grafted ILs to allow only one type of ions enter carbon pores is applied for an in-depth understanding of the charging mechanism. Interestingly, we have found that this strategy can be also used to tune the charge distribution of in-pore ions, and thus can optimize the potential window of individual electrode to increase the energy density of supercapacitor. Now we are still working on this, and the relative results will be reported soon.

Response to reviewer 4:

Reviewer’s summary remark: “The manuscript by Dou et al. presents an interesting novel way to separate the contributions of anions and cations in the charging mechanism of supercapacitor electrodes. The key advance here is using ionic solutions where one type of ion is specifically designed to be too large to fit inside the micropores of the electrode. As the authors acknowledge (ref 39), studies have been carried out previously on electrolytes with large ion size differences and similar observations have been reported. However, this work represents (to my knowledge) the first instance where this phenomenon has been exploited in a systematic way to study the charging mechanism. Experimental methods for gaining such information about the behavior of ions in AC electrodes during charging are still not common and this work has provided some important fundamental insight, and is an important contribution to the supercapacitor field. I therefore believe it should be published in Nature Communications subject to addressing a few comments:”

Reply: Thank you very much for your careful review on our manuscript as well as the highly positive evaluation on our work. We also thank for your constructive comments. The manuscript has been revised carefully accordingly. The detailed answers are listed as follows.

Comment 1: “There are several mistakes or use of unclear English/grammar throughout and I recommend that this paper is proof-read by a native speaker prior to publication.

Reply: Thank you very much for your constructive comment, Inspired by your comment, this manuscript has been polished by an English language editing service.

Figure R1 shows the certificate of English editing.

EnPapers.Com **CERTIFICATE OF ENGLISH EDITING**

To whom it may concern:

This memo is to certify that the paper titled Silica-grafted ionic liquids: charging behaviors of cations and anions in supercapacitors has been edited for language by EnPapers, a company dedicated to helping international researchers publish their findings in the best English language journals possible.

Our International paper editing service is performed by a subject expert editor and approved by two senior editors. All our editors are native English speakers.

The certificate is being issued upon the request of the client. If you have any questions, please contact papers@enpapers.com

Signature of the editor representative:

Martin J. Booth

Figure R1. Certificate of English editing.

Comment 2: In particular, I find the description of the “assembled” CV curves and “fitting” curves quite unclear and it took me a few re-reads to understand what they meant. I would suggest referring to “summed” CV curves or similar.

Reply: Thank you so much for this comment. The inappropriate “*assembled CV*” and “*fitting CV*” were modified as the “*summed CV*” in our revised manuscript.

Comment 3: The authors state “In addition, it can be found that the smaller capacitance of the fitting curves than those of the conventional electrolytes when the direction of the scan rate is reversed.” I cannot see what this is referring to, and the term “direction of scan rate” is scientifically meaningless. This part needs to be

explained better.”

Reply: According to your suggestion, the initial sentence has been modified as “*The capacitance of the summed curves was smaller than that of the conventional electrolytes upon the reversal of the potential.*”

Comment 4: “I agree with reviewer 1’s point about the authors’ description of the limitations of other techniques and I still feel that the introduction dwells more than necessary on the drawbacks of other methods which have contributed greatly to the current understanding. The method proposed in the current work also has its drawbacks - it seems rather laborious and complicated, and does not provide molecular-level detail of other spectroscopic techniques. I think a better way to frame the current work would be to focus on the complementary and additional knowledge it brings to the other methods that have also provided great insight.”

Reply: Thank you very much for your constructive comment. Inspired by your comment, we have realized that it is unnecessary to introduce the limitations of other techniques in detail. Thus, the corresponding contents have been deleted in the introduction section of our revised manuscript. We found that, after this revision, the novelty description of our work is more accurate.

Comment 5: I do not agree with reviewer 1’s criticism of the fact that the work does not examine a pure IL. While it is true that ILs are of great interest as supercapacitor electrolytes, solvated electrolytes are more widely used and remain poorly understood. I do not see that it is “better” to study ILs. The main aim of the manuscript (and of much of the wider supercapacitor community) is simply to gain a better understanding of the behavior of ions at electrochemical interfaces, which still remains a great challenge. In some ways solvated electrolytes are even more complex than ILs with solvation/desolvation phenomena potentially contributing to the energy storage process, and so I do not see this as a negative aspect of the work.

Reply: I appreciate deeply for your accurate evaluation on our work.